

# Comparative performance of twelve machine learning models in predicting COVID-19 mortality risk in children: a population-based retrospective cohort study in Brazil

Adriano Lages dos Santos[1,2], Maria Christina L. Oliveira[2], Enrico A. Colosimo[3], Robert H. Mak[4], Clara C. Pinhati[2], Stella C. Gallante[2], Hercílio Martelli-Júnior[5], Ana Cristina Simões e Silva[2] and Eduardo A. Oliveira[2]

[1] Engineering and Informatics, Federal Institute of Science and Technology of Minas Gerais, Belo Horizonte, Minas Gerais, Brazil
[2] Department of Pediatrics, School of Medicine, Federal University of Minas Gerais, Belo Horizonte, Minas Gerais, Brazil
[3] Department of Statistics, Universidade Federal de Minas Gerais, Belo Horizonte, Minas Gerais, Brazil
[4] Division of Pediatric Nephrology, Rady Children's Hospital, University of California, San Diego, San Diego, California, United States
[5] Department of Health Sciences, School of Odontology, Montes Claros State University, Montes Claros, Minas Gerais, Brazil

## ABSTRACT

The COVID-19 pandemic has catalyzed the application of advanced digital technologies such as artificial intelligence (AI) to predict mortality in adult patients. However, the development of machine learning (ML) models for predicting outcomes in children and adolescents with COVID-19 remains limited. This study aimed to evaluate the performance of multiple machine learning models in forecasting mortality among hospitalized pediatric COVID-19 patients. In this cohort study, we used the SIVEP-Gripe dataset, a public resource maintained by the Ministry of Health, to track severe acute respiratory syndrome (SARS) in Brazil. To create subsets for training and testing the machine learning (ML) models, we divided the primary dataset into three parts. Using these subsets, we developed and trained 12 ML algorithms to predict the outcomes. We assessed the performance of these models using various metrics such as accuracy, precision, sensitivity, recall, and area under the receiver operating characteristic curve (AUC).

Among the 37 variables examined, 24 were found to be potential indicators of mortality, as determined by the chi-square test of independence. The Logistic Regression (LR) algorithm achieved the highest performance, with an accuracy of 92.5% and an AUC of 80.1%, on the optimized dataset. Gradient boosting classifier (GBC) and AdaBoost (ADA), closely followed the LR algorithm, producing similar results. Our study also revealed that baseline reduced oxygen saturation, presence of comorbidities, and older age were the most relevant factors in predicting mortality in children and adolescents hospitalized with SARS-CoV-2 infection. The use of ML models can be an asset in making clinical decisions and implementing evidence-based patient management strategies, which can enhance patient outcomes

Corresponding author
Adriano Lages dos Santos,
adrianocomp@gmail.com

and overall quality of medical care. LR, GBC, and ADA models have demonstrated efficiency in accurately predicting mortality in COVID-19 pediatric patients.

# INTRODUCTION

Since the onset of the COVID-19 pandemic, the global community has witnessed remarkable progress in artificial intelligence (AI), particularly in machine learning (ML) algorithms such as large language models (LLMs) (*Thirunavukarasu et al., 2023*; *Yu et al., 2023*). These models have played a crucial role in assisting researchers globally in devising innovative solutions to the diverse challenges in the healthcare field (*Howell, Corrado & DeSalvo, 2024*; *Li et al., 2024*; *Zhang et al., 2023*). The utilization of generative AI to provide diagnoses and prognoses for various diseases across different medical specialties has experienced substantial growth in recent years (*Bohr & Memarzadeh, 2020*; *Elias et al., 2024*; *Gurcan, 2025*; *Jain et al., 2024*; *Yip et al., 2023*).

AI has been rapidly and extensively implemented in routine clinical care, encompassing enhanced prognosis and diagnosis, robot-assisted surgery, rehabilitation, data science, and precision medicine, all of which have benefited from advancements in computer hardware and big data development. Numerous studies have been conducted utilizing AI tools to predict various outcomes in different medical domains using diverse types of data, such as text and images (*Buch, Ahmed & Maruthappu, 2018*; *Ching et al., 2018*; *Li et al., 2024*; *Xu et al., 2023*). The COVID-19 pandemic has accelerated the adoption of AI and big-data technologies in healthcare, epidemiology, and public health. With COVID-19 impacting communities in different ways, research has increasingly turned to big data analytics and AI tools to track and monitor the spread of the virus and its effects on public health and the global economy (*Galetsi, Katsaliaki & Kumar, 2022*; *Sipior, 2020*). These technologies have played a crucial role in understanding, managing, and mitigating the impact of the virus by addressing the various challenges posed by the pandemic, including diagnosis, treatment, and vaccine development (*Roosli, Rice & Hernandez-Boussard, 2021*).

More than four years have elapsed since the initial declaration of the COVID-19 pandemic. According to the World Health Organization (WHO), as of February 2024, the global count of confirmed COVID-19 cases has surpassed 826 million, with COVID-19-related fatalities reaching 7 million worldwide (https://data.who.int/dashboards/covid19/cases?n=c).

Approximately 10–20% of confirmed SARS-CoV-2 infections and less than 0.5% of fatalities occur in people under 18 years of age (*Silverberg et al., 2024*; *Swann et al., 2020*). While the virus generally results in less severe illness and fewer deaths among children and adolescents than among adults, some pediatric cases still lead to serious outcomes (*Howard-Jones et al., 2022*). Importantly, a comprehensive study found that over 90% of COVID-19-related deaths in young individuals were reported in low-middle-income

countries (LMICs) (*Kitano et al., 2021*). In this meta-analysis that included a cohort of 3,379,049 children, *Kitano et al. (2021)* reported an estimated case fatality rate of 0.29% (95%CI, [0.28–0.31%]) in LMIC, which was significantly higher than that reported in high-income countries (HIC) (0.03% [0.03–0.03%]). Consistent with these findings, we have demonstrated in a series of studies since the onset of the pandemic that the mortality rate has increased to 7.9% among children hospitalized for COVID-19 in Brazil. Furthermore, using conventional statistical methodologies, we have demonstrated increased mortality and a significant impact of the social determinants of health and ethnicity on COVID-19 outcomes in Brazil's pediatric population (*Oliveira et al., 2021*, *2023a*, *2024*, *2022*). Additional studies from developing countries have corroborated the trend of unfavorable outcomes among hospitalized children in LMICs (*Nachega et al., 2022*). In contrast, data from HIC exhibited substantially more favorable outcomes in hospitalized children, with mortality rates ranging from 0.1% (*Swann et al., 2020*) to 0.4% (*Doenhardt et al., 2024*), which was predominantly observed in patients with complex diseases and comorbidities.

These observations underscore the significance of acquiring comprehensive data on COVID-19 clinical outcomes in young individuals from LMICs to accurately assess the overall impact of the disease on pediatric patients. However, there has been a paucity of population-based studies examining COVID-19 incidence and outcomes in children and adolescents, particularly in LMICs (*Rankin et al., 2021*).

As the COVID-19 pandemic wanes, it is crucial to accurately quantify its impact on children and adolescents to develop targeted prevention strategies against future public health threats for this vulnerable group (*Chiotos & Fitzgerald, 2023*). Nevertheless, a comprehensive understanding of the complex interplay between individual factors and social inequities in shaping the outcomes of COVID-19 in children and adolescents remains to be fully elucidated (*Oliveira, Colosimo & Simoes, 2022*). In this context, AI techniques can provide valuable insights into decision-making processes, including the development of effective public health policies with the ultimate aim of reducing morbidity and mortality. Although ML algorithms have been widely applied to diagnose and predict COVID-19 outcomes in adults, their utilization in pediatric populations remains underexplored (*Dos Santos et al., 2024*).

Nevertheless, a significant gap remains regarding the utilization of AI tools in pediatrics. For instance, in a recent systematic review, we demonstrated that within the context of the COVID-19 pandemic, there has been a notable scarcity of studies on the development of clinical models for predicting outcomes in children and adolescents using AI algorithms, particularly when compared to the literature encompassing adult patients. Furthermore, our study revealed the substantial limitations of pediatric studies in this domain, including insufficient sample sizes, inconsistent reporting methodologies, biases in data sources, and ethical considerations (*Dos Santos et al., 2024*).

The subsequent sections of this article are structured as follows. "Related Work" presents a review of previous research on AI technologies in pediatrics. "Materials and Methods" elucidates the study's methodology in detail. The "Results" section presents the main findings of the study, and the "Discussion" section provides a comprehensive

analysis of our findings in relation to the existing literature. The key findings and future research directions are summarized in "Conclusions."

## RELATED WORK

This section reviews and presents the literature on predicting mortality and other outcomes in children and adolescents (including COVID-19-related deaths) using machine learning techniques.

*Zhang, Xiao & Luo (2023)* employed ML techniques to estimate infant mortality rates in the United States, considering various factors related to birth facilities, prenatal care, labor and delivery conditions, and neonatal characteristics. This study analyzed data spanning from 2016 to 2021, encompassing 116,309 infant deaths out of 22,669,736 live births. Among the five ML models evaluated, XGBoost demonstrated the highest predictive accuracy, achieving an area under the curve (AUC) of 93% and an average precision (AP) score of 0.55. The findings underscored the advantages of utilizing the original imbalanced dataset rather than artificially balanced datasets generated through oversampling techniques, as the former produced superior predictive outcomes. Furthermore, the model's validation using data from 2020 to 2021 confirmed its robustness, maintaining an AUC of 93% and AP score of 0.52. The model's consistent performance across both the pre-pandemic (2016–2019) and pandemic (2020–2021) periods suggests its potential utility in shaping public health strategies aimed at reducing infant mortality rates.

A study conducted by *Byeon (2022)* employed a population-based cross-sectional survey to assess the impact of the COVID-19 pandemic on the prevalence of obesity among adolescents in South Korea. To develop a predictive model for adolescent obesity, researchers have utilized categorical boosting, specifically the CatBoost algorithm. The performance of the model was assessed using multiple evaluation metrics, revealing an AUC of 68% and an overall accuracy of 82%. The analysis incorporated various factors, including physical activity level, academic performance, and lifestyle habits, to identify potential risk factors associated with adolescent obesity. This study's methodological rigor, demonstrated through the use of the CatBoost algorithm and comprehensive performance evaluation, highlights its contribution to understanding obesity risk among South Korean adolescents in the post-pandemic context.

*Gao et al. (2022)* proposed a hybrid approach that integrates domain knowledge-based features with data-driven methodologies to predict pediatric COVID-19 hospitalization and disease severity. The study utilized two cohorts, which were divided into training, validation, and testing sets at a 6:1:3 ratio. The training set was used for model fitting, the validation set for hyperparameter tuning, and the testing set for performance evaluation. The models were assessed using the area under the receiver operating characteristic curve (AUROC), area under the precision-recall curve (AUPRC), and minimum value between recall and precision (Min [Re, Pr]). The best-performing model, MedML, demonstrated a 3% improvement in AUROC and a 4% increase in AUPRC for predicting hospitalization. For severity prediction, it outperformed the best baseline model by 7% in the AUROC and 14% in the AUPRC. The researchers employed N3C Data Enclave with Code Workbook and mini-batch gradient descent for model training, setting the batch size to 128. The

findings indicate that MedML maintains generalizability across all nine national geographical regions of the United States and remains robust throughout the different pandemic phases. The authors highlighted MedML's role as a bridge between clinicians, data engineers, and computer scientists, enhancing clinical decision-making through intuitive knowledge representation, explainable model construction, and advanced computational capabilities.

*Pavliuk & Kolesnyk (2023)* developed an ML model to analyze and predict hospitalization rates among children in the Lviv region during the fourth wave of the COVID-19 pandemic, marked by the predominance of the Omicron variant. The increase in pediatric hospitalizations was primarily attributed to the children's high levels of social interaction and the low vaccination rates observed in Ukraine. Utilizing publicly available data, the proposed ML model consists of two main components: data analysis and predictive modeling. The Pearson correlation coefficient was applied to assess hospitalization trends, whereas neural networks were employed to generate short- and medium-term forecasts. *Mamlook et al. (2021)* conducted a comparative evaluation of five widely recognized ML techniques: artificial neural networks (ANN), random forests (RF), support vector machines (SVM), decision trees (DT), and gradient boosted trees (GBM) for detecting COVID-19 in pediatric patients. Model performance was assessed using a 10-fold cross-validation procedure. The results indicated that the classification and regression tree (CART) model outperformed other approaches, achieving an accuracy of 92.5% for binary classification (positive *vs.* negative) based on laboratory test results. Key predictors, including leukocyte count, monocyte levels, potassium concentration, and eosinophil count, were identified as significant factors for COVID-19 detection. This study underscores the potential of ML-based models as valuable tools for healthcare professionals, aiding in the prediction of COVID-19 in children and reinforcing laboratory diagnostic findings.

*Ma et al. (2021)* explored whether clinical symptoms and laboratory test results could serve as reliable predictors for determining the necessity of computed tomography (CT) scans in pediatric patients with positive RT-PCR results. Data from 244 pediatric cases were analyzed using advanced decision-tree-based ML models. The study identified age, lymphocyte count, neutrophil levels, ferritin concentration, and C-reactive protein levels as critical indicators for predicting CT scan outcomes. The developed decision support system demonstrated robust predictive performance, achieving an AUC of 84%, with an accuracy of 82% and a sensitivity of 84%. These findings suggest the potential for reassessing the routine use of CT imaging in pediatric COVID-19 cases, emphasizing that, in some instances, alternative diagnostic approaches may suffice.

*Nugawela et al. (2022)* developed a predictive model to identify children and adolescents at an increased risk of developing long COVID, defined as the presence of at least one persistent symptom impairing daily activities 3 months after a positive SARS-CoV-2 RT-PCR test. The study utilized data from a nationally matched cohort of individuals aged 11–17 years, including both SARS-CoV-2-positive and test-negative participants. The model incorporated a range of predictive factors, including SARS-CoV-2 infection

status, demographic characteristics, quality of life and functional status, physical and mental health indicators, levels of loneliness, and number of symptoms reported at the time of testing. Using logistic regression, the model achieved an accuracy of 83%, demonstrating its strong calibration and discrimination capabilities. These findings highlight the potential of predictive modeling in identifying vulnerable pediatric populations and informing targeted interventions to mitigate the long-term impact of COVID-19. This study aimed to evaluate the potential of machine learning (ML) models to predict mortality and hospital discharge among hospitalized children and adolescents with laboratory-confirmed COVID-19. Using a comprehensive nationwide dataset provided by the Brazilian government, we sought to identify the most critical predictors for these models and to understand their decision-making processes. Additionally, we assessed the effectiveness of the model in forecasting COVID-19-related deaths.

## MATERIALS AND METHODS

### Study design and dataset description

This retrospective cohort study used data from the Surveillance Information System for Influenza (SIVEP-Gripe) to examine COVID-19 hospitalizations among individuals under 18 years of age in Brazil. Established in 2009 by the Ministry of Health, the SIVEP-Gripe is a nationwide database that captures data on severe acute respiratory infections. Since the COVID-19 pandemic, it has been the primary source of hospitalization data. Mandatory reporting from both public and private hospitals ensures comprehensive coverage. The database contains the demographic and clinical information of all hospitalized patients. We analyzed data from epidemiological week 08, 2020, to week 08, 2023, encompassing individuals aged under 18 years with confirmed SARS-CoV-2 infection *via* RT-qPCR or antigen testing upon hospital admission.

### Data preparation

Over the designated period, 56,330 patient records with verified RT-PCR test outcomes for SARS-CoV-2 infection were documented. After completing the required procedures for preprocessing data for the machine learning algorithms, 24,097 records were chosen for the training, validation, and testing stages of the models. The subset of data from the SIVEP-Gripe dataset, which includes information about children and adolescents, is hereafter referred to as the SIVEP-Kids dataset.

In the SIVEP-Kids dataset, there are 37 primary features in four main categories: patient demographics (four features), clinical features (12 features), personal disease/comorbidity history (14 features), virus strain information (one feature), vaccine information (two features), a feature indicating the number of different comorbidities a patient has, a feature indicating whether a patient has comorbidities or not, a feature categorizing the number of comorbidities a patient has, a feature indicating the time of the outcome, and an output variable (0: survived and 1: deceased) for COVID-19 patients. The primary features of the SIVEP-Kids dataset are presented in Table 1.

**Table 1 Primary features documented in the SIVEP-Kids database.**

| No. | Feature name | Variable type | No. | Feature name | Variable type |
|-----|--------------|---------------|-----|--------------|---------------|
| 1 | Gender | Nominal | 21 | Hypertension | Nominal |
| 2 | Age | Numeric | 22 | Immunosuppression | Nominal |
| 3 | Ethnicity | Nominal | 23 | Renal disease | Nominal |
| 4 | Region | Nominal | 24 | Asthma | Nominal |
| 5 | Virus strain | Nominal | 25 | Hematology disease | Nominal |
| 6 | Dyspnea | Nominal | 26 | Neurology | Nominal |
| 7 | Fever | Nominal | 27 | Oncology | Nominal |
| 8 | Cough | Nominal | 28 | Transplanted | Nominal |
| 9 | Odynophagia | Nominal | 29 | Down syndrome | Nominal |
| 10 | Diarrhea | Nominal | 30 | Other syndrome | Nominal |
| 11 | Vomit | Nominal | 31 | Nosocomial | Nominal |
| 12 | Abdominal pain | Nominal | 32 | Comorbidities dichotomic | Nominal |
| 13 | Ageusia | Nominal | 33 | Total comorbidities | Numeric |
| 14 | Anosmia | Nominal | 34 | Number of vaccine doses | Numeric |
| 15 | Respiratory distress | Nominal | 35 | Comorbidities categoric | Nominal |
| 16 | Oxygen saturation reduced | Nominal | 36 | Time for outcome | Numeric |
| 17 | Diabetes | Nominal | 37 | Vaccinated | Nominal |
| 18 | Obesity | Nominal | 38 | Outcome (Target Variable) | Nominal |
| 19 | Cardiology | Nominal | | | |
| 20 | Pulmonary | Nominal | | | |

Regarding the primary features presented in the SIVEP-Kids dataset, the ethnicity feature had five categories: Asian, Black, Brown, Indigenous, and White. Similarly, the region was divided into five regions: Central West, North, Northeast, South, and Southeast. The virus strain feature identified four types of strains in the dataset: ancestral, delta, gamma, and omicron. For features 6 through 32, all are of the nominal type and have values of "Yes" or "No," indicating the presence or absence of a specific disease or clinical condition in the patient. The total comorbidity feature records the total number of comorbidities per patient in the SIVEP-Kids dataset. Feature 34 (number of vaccine doses) had valid values ranging from zero to three doses. Feature 38 is the target variable of this study, with three types of outcomes: discharge, death, and in-hospital, with the latter referring to cases in which the patient is still in the hospital in an ongoing clinical situation. In the present study, we considered only two types of outcomes in the target variable: death and discharge. This decision aimed to enhance the accuracy of machine learning algorithms, as multi-class problems (those with more than two classes in the target variable) are challenging and tend to reduce the accuracy of ML models because of the large number of decision boundaries to navigate, often failing to accurately separate instances across more than two classes (*Bengio, Weston & Grangier, 2010*; *Del Moral, Nowaczyk & Pasham, 2022*). Detailed information on the clinical, demographic, and epidemiological covariates recorded in the SIVEP-Gripe is described elsewhere (*Oliveira et al., 2021*, *2023b*, *2024*, *2022*).

## Data pre-processing

Data preprocessing is a critical step in addressing the influence of irrelevant, redundant, and unreliable data, ultimately improving data quality and resolving inconsistencies (*Garcia, Luengo & Herrera, 2105*). In this study, data preprocessing was conducted prior to training the machine learning models. Initially, the patient records with missing data were removed from the dataset. For example, records of sex, ethnicity, and reduced oxygen saturation were excluded if any missing values were detected. Missing values for the target variable were treated as the absence of the outcome of interest (death). Additionally, we utilized categorical encoding to transform nominal data into numerical representations. By applying one-hot encoding, we ensured that our analysis was guided by intrinsic relationships within the data rather than by the constraints of non-numerical representations (*Xiang et al., 2021*).

After applying the criteria for excluding data in the pre-processing step, we obtained a final sample consisting of 24,097 records. The dataset comprised 22,586 and 1,511 cases in the discharge and death classes, respectively. An imbalanced input distribution can lead to a bias in the results towards the dominant class, potentially skewing model performance and reducing generalizability. To address the problem posed by an imbalanced dataset, we employed the Synthetic Minority Over-sampling Technique (SMOTE) method, as outlined in https://imbalanced-learn.org/stable/. The SMOTE algorithm, which is widely utilized for synthetic oversampling, generates artificial samples for the minority class by randomly selecting instances from the minority class and their k-nearest neighbors. In this approach, a random data instance along with its k-nearest neighbors is chosen. Subsequently, the second data instance was selected from this set of k-nearest neighbors (*Dorn et al., 2021*). The synthesis of a new sample occurred along the line connecting these two instances as a convex combination. This process was iterated until a balance was achieved between minority and majority classes. The SMOTE method mitigates the risk of overfitting, distinguishing it from the random oversampling technique, and it is recognized for its potential to produce better results (*Erol et al., 2022*; *Wang et al., 2021*; *Wongvorachan, He & Bulut, 2023*).

## Feature selection

Chi-square tests were used to discern statistically significant differences between the outcomes of discharged and deceased patients. Feature importance scores derived from XGBoost and random forests (as detailed in Fig. S1) were utilized to identify the essential variables for forecasting COVID-19 mortality. This methodology aims to increase the interpretability and steadfastness of mortality prediction models.

Feature selection techniques exhibited elevated scores for robust predictors such as overall comorbidities, diminished oxygen saturation, and age. Nevertheless, some disparities were evident in the importance scores between XGBoost and random forest for specific parameters. XGBoost showed considerable importance in reducing oxygen saturation and overall comorbidities, whereas random forest allocated minimal importance. A statistically significant difference ($P < 0.01$) in oxygen saturation and total comorbidities was observed between patients who survived and those who died.

Chi-square tests were applied to recognize crucial mortality predictors, demonstrating moderate to high importance in XGBoost and low importance in random forest.

Owing to the inconsistencies observed between the two methods, we opted to select the most pertinent features for training the models using the chi-squared test. Consequently, we developed three distinct datasets to train and validate the machine learning models. These datasets included a dataset with features selected using the chi-squared test, a dataset with features chosen by two pediatricians, and a dataset with all 37 features, according to Table 1, except for the target variable. Our objective was to determine the dataset that yielded the most favorable results.

The dataset containing characteristics chosen by pediatricians comprised 17 features: sex, age, ethnicity, region, virus strain, dyspnea, fever, cough, odynophagia, abdominal pain, ageusia, anosmia, respiratory distress, reduced oxygen saturation, total comorbidities, vaccine doses, and nosocomial. The dataset selected by the chi-squared test comprised 24 features: age, ethnicity, region, viral strain, dyspnea, cough, respiratory distress, reduced oxygen saturation, cardiology, pulmonary disease, hypertension, immunosuppression, renal disease, asthma, total comorbidities, comorbidities, dichotomous comorbidities, time for outcome, vaccine doses, hematology, neurology, oncology, Down syndrome, and nosocomial infection.

For the purpose to conducting feature selection calculations using the chi-square test, XGBoost, and random forest, the Scikit-learn library in its version 1.3.1 was used. The Pycaret library version 3.1.0 was employed for training and validating the models. Statistical significance was set at $P < 0.01$.

## Outcomes

The primary endpoint was COVID-19-related death. Additionally, we assessed the severity of the disease, including hospitalization, need for respiratory support (none, non-invasive oxygen support, and mechanical ventilation), and admission to the intensive care unit (ICU).

## Model development

In this study, a total of twelve machine learning algorithms were employed to develop predictive models. These algorithms included gradient boosting (GB), AdaBoost (ADA), CatBoost (Cat), random forest (RF), extreme gradient boosting (XGBoost), extra trees (ET), logistic regression (LR), linear discriminant analysis (LDA), decision tree (DT), naïve Bayes (NB), k-nearest neighbors (KNN), and Quadratic Discriminant Analysis (QDA) (*Dorn et al., 2021*).

These models were selected due to their superior performance compared to deep learning algorithms. Recent studies indicate that for tabular data, ML algorithms such as XGBoost, CatBoost, logistic regression, and decision tree family algorithms exhibit better performance than neural networks. Notwithstanding ongoing research efforts, neural networks have demonstrated limited efficacy in the processing of tabular data (*Hwanga & Jongwoo, 2023*; *Shmuel, Glickman & Lazebnik, 2024*; *Shwartz-Ziv & Armon, 2022*; *Sivapathasundaram & Poravi, 2021*).

**Table 2 The hyperparameters of the selected ML algorithms for COVID-19 mortality prediction in children and adolescents.**

| ML algorithms | Hyperparameters used to create the models |
|---|---|
| GBC | criterion='friedman_mse', learning_rate=0.0005, max_depth=9, max_features='log2', min_impurity_decrease=0.001, min_samples_leaf=1, min_samples_split=9, n_estimators=120, subsample=0.9, tol=0.0001, validation_fraction=0.1. |
| ADA | algorithm='SAMME', learning_rate=0.005, n_estimators=260. |
| CATBOOST | Iterations=1000, learning_rate=0.1, depth=6, l2_leaf_reg=3.0, subsample=0.8, colsample_bylevel=0.8, border_count=128, loss='log_loss'. |
| RF | criterion='gini', max_depth=4, max_features=1.0, max_leaf_nodes=None, min_impurity_decrease=0.3, min_samples_leaf=2, min_samples_split=7, n_estimators=90. |
| XGBOOST | booster='gbtree', colsample_bytree=1, learning_rate=0.4, max_depth=1, min_child_weight=2, n_estimators=120, objective='binary: logistic' |
| ET | criterion='gini', max_depth=4, max_features=1.0, min_impurity_decrease=0.3, min_samples_leaf=2, min_samples_split=7, n_estimators=90. |
| LR | C=0.662, fit_intercept=True, intercept_scaling=1, l1_ratio=None, max_iter=1000, penalty='l2', solver='lbfgs', tol=0.0001. |
| LDA | shrinkage=0.4, solver='lsqr', tol=0.0001. |
| DT | criterion='entropy', max_depth=4, max_features=1.0, min_impurity_decrease=0.5, min_samples_leaf=3, min_samples_split=2, splitter='best'. |
| NB | var_smoothing=1 |
| KNN | leaf_size=30, metric='manhattan', n_neighbors=50, p=2, weights='distance'. |
| QDA | reg_param=0.29, tol=0.0001. |

The evaluation process involved the use of k-fold cross-validation, which is known to have low bias and variation. The optimized hyperparameters for the machine learning algorithms are provided in Table 2, with constant values maintained across the three variations of the SIVEP-Kids dataset.

## Assessment metrics

The performance of the predictive model was evaluated using various metrics, such as accuracy, precision, sensitivity, F1 score, and area under the ROC curve (AUC). A comprehensive analysis was conducted across all 12 machine learning algorithms to determine the best model for predicting mortality in COVID-19 patients (*Fawcett, 2006*; *Powers, 2011*; *Sokolova & Lapalme, 2009*). Other performance metrics information will be detailed in the results section.

Accuracy: Measures the proportion of correctly classified instances.

$$Accuracy = \frac{TP + TN}{TP + TN + FP + FN}$$

where $TP$ = true positives, $TN$ = true negatives, $FP$ = false positives, and $FN$ = false negatives (*Sokolova & Lapalme, 2009*)

Precision: Assesses how many of the predicted positive instances are actually correct.

$$Precision = \frac{TP}{TP + FP}.$$

Precision is particularly important in applications where false positives must be minimized (*Powers, 2011*)

Recall (Sensitivity): Measures the proportion of actual positives that were correctly identified.

$$Recall = \frac{TP}{TP + FN}.$$

This metric is crucial in applications where false negatives are costly (*e.g.*, medical diagnosis) (*Sokolova & Lapalme, 2009*).

F1-score: A harmonic mean of precision and recall, balancing both metrics.

$$F1 = 2 \times \frac{Precision \times Recall}{Precision + Recall}.$$

F1-score is useful when dealing with imbalanced datasets.

AUC-ROC: Measures the ability of the model to distinguish between classes by plotting the true positive rate against the false positive rate.

$$AUC = \int_{\{0}^{1} TPR(FPR)d(FPR)$$

where:

*FPR*–false positive rate

*TPR*–true positive rate

*d (FPR)* represents the infinitesimal variation in the false positive rate (FPR).

In practice, AUC is numerically calculated as the sum of the areas under the ROC curve, approximating the integral by summing small rectangular or trapezoidal regions along the curve.

AUC-ROC is widely used in binary classification problems to assess model discrimination power (*Fawcett, 2006*).

We employed SHAP summary and force plots to elucidate the decision-making processes of the models. Given its superior performance across all datasets, the gradient boosting classifier (GBC) was selected for in-depth analysis. SHAP summary plots visualize feature importance by mapping the impact of each feature on the model output to a dot on the horizontal axis. The position of the dot represents the SHAP value, which quantifies the contribution of the feature to the prediction. Feature values were color-coded (red: high, blue: low) to reveal the direction and magnitude of their influence. For detailed individual predictions, please refer to the force plots in the Supplemental Material.

## RESULTS

### Feature selection

Twenty-four features, comprising demographic and clinical factors, were identified as the most relevant predictors using the chi-square independence test (Table 3). Additionally,

**Table 3 The significance levels, importance scores, and mean decreases in Gini for the key variables in COVID-19 mortality prediction were computed using the XGBoost, random forest, and chi-squared tests.**

| N° | Feature name | Chi-squared test | | Random forest | XGBoost |
|---|---|---|---|---|---|
| | | X² | P-value | Mean decrease impurity | Importance score |
| 1 | Age | 396.94 | <0.001 | 0.171 | 0.029 |
| 2 | Region | 17.02 | <0.001 | 0.084 | 0.035 |
| 3 | Ethnicity | 9.59 | <0.001 | 0.04 | 0.022 |
| 3 | Virus Strain | 34.25 | <0.001 | 0.048 | 0.023 |
| 4 | Dyspnea | 57.69 | <0.001 | 0.026 | 0.025 |
| 5 | Cough | 37.89 | <0.001 | 0.030 | 0.050 |
| 6 | Respiratoy distress | 79.53 | <0.001 | 0.025 | 0.035 |
| 7 | Oxygen saturation reduced at admission | 175.43 | <0.001 | 0.027 | 0.125 |
| 8 | Obesity | 66.27 | <0.001 | 0.005 | 0.020 |
| 9 | Cardiology | 212.09 | <0.001 | 0.009 | 0.029 |
| 10 | Pulmonary | 33.75 | <0.001 | 0.006 | 0.025 |
| 11 | Hypertension | 25.17 | <0.001 | 0.002 | 0.011 |
| 12 | Immunosuppression | 108.01 | <0.001 | 0.008 | 0.032 |
| 13 | Renal | 49.48 | <0.001 | 0.004 | 0.016 |
| 14 | Asthma | 18.74 | <0.001 | 0.007 | 0.040 |
| 15 | Total comorbidities | 861.55 | <0.001 | 0.021 | 0.106 |
| 16 | Comorbidities dichotomic | 527.13 | <0.001 | 0.012 | 0.000[a] |
| 17 | Comorbidities categoric | 830.74 | <0.001 | 0.019 | 0.000[a] |
| 18 | Time for outcome | 504.48 | <0.001 | 0.208 | 0.023 |
| 19 | Hematology | 27.33 | <0.001 | 0.004 | 0.014 |
| 20 | Neurology | 278.64 | <0.001 | 0.013 | 0.024 |
| 21 | Oncology | 52.08 | <0.001 | 0.003 | 0.020 |
| 22 | Down syndrome | 79.18 | <0.001 | 0.006 | 0.023 |
| 23 | Nosocomial | 70.17 | <0.001 | 0.013 | 0.024 |

Note:
[a] Two (comorbidities dichotomic and comorbidities categoric) had zero values for importance scores calculated with XGBoost. This is because the XGBoost algorithm detected multicollinearity between the two characteristics and total comorbidities. In this case, the two columns are ignored by the algorithm.

Table 3 shows mean decreases in impurity and the importance scores of these variables calculated using the XGBoost and random forest algorithms. The descriptive statistics of these features are summarized in Table 4.

Table 3 indicates that age, cardiovascular disease, decreased oxygen saturation upon admission, total comorbidities, and time to outcome were significantly associated with patient outcomes, as determined using the chi-square test. These factors demonstrated a strong predictive power in distinguishing between fatal and discharged cases. This statistical significance is also apparent in the developed models and was of paramount importance in the training process.

In contrast, odynophagia, abdominal pain, fever, vaccination, transplant, diabetes mellitus, vomiting, other syndromes, sex, diarrhea, and ageusia were less predictive of COVID-19 mortality. Despite their clinical importance in treatment and mortality risk

**Table 4 Descriptive statistics of the most important variables selected in the feature selection phase for mortality in COVID-19 children and adolescents' patients.**

| N° | Feature name | Variable type | Frequency or mean ± SD |
|---|---|---|---|
| 1 | Age | Numeric | 5.04 ± 5.25 |
| 2 | Region | Nominal | Southeast (10,819) |
|  |  |  | South (4,379) |
|  |  |  | Northeast (4,033) |
|  |  |  | North (2,609) |
|  |  |  | Central West (2,257) |
| 3. | Ethnicity | Nominal | Asian (178) |
|  |  |  | Black (778) |
|  |  |  | Brown (11,467) |
|  |  |  | Indigenous (221) |
|  |  |  | White (11,453) |
| 4 | Virus Strain | Nominal | Omicron (13,432) |
|  |  |  | Gamma (8,251) |
|  |  |  | Delta (2,414) |
| 5 | Dyspnea | Nominal | Haven't (11,126) |
|  |  |  | Have (12,971) |
| 6 | Cough | Nominal | Haven't (7,198) |
|  |  |  | Have (16,899) |
| 7 | Respiratory distress | Nominal | Haven't (11,245) |
|  |  |  | Have (12,852) |
| 8 | Oxygen saturation reduced at admission | Nominal | Haven't (12,018) |
|  |  |  | Have (12,079) |
| 9 | Obesity | Nominal | Haven't (23,675) |
|  |  |  | Have (422) |
| 10 | Cardiology | Nominal | Haven't (23,314) |
|  |  |  | Have (783) |
| 11 | Pulmonary | Nominal | Haven't (23,608) |
|  |  |  | Have (489) |
| 12 | Hypertension | Nominal | Haven't (24,029) |
|  |  |  | Have (68) |
| 13 | Immunosuppression | Nominal | Haven't (23,580) |
|  |  |  | Have (517) |
| 14 | Renal | Nominal | Haven't (23,876) |
|  |  |  | Have (221) |
| 15 | Asthma | Nominal | Haven't (22,711) |
|  |  |  | Have (1,386) |
| 16 | Total comorbidities | Numeric | 0.22 ± 0.54 |
|  |  |  | (0, 19,670) |
|  |  |  | (1, 3,512) |
|  |  |  | (2, 757) |
|  |  |  | (3, 128) |

(Continued)

| N° | Feature name | Variable type | Frequency or mean ± SD |
|----|--------------|---------------|------------------------|
| | | | (4, 24) |
| | | | (5, 2) |
| | | | (6, 2) |
| | | | (7, 1) |
| | | | (10, 1) |
| 17 | Comorbidities dichotomic | Nominal | Haven't (19,670) |
| | | | Have (4,427) |
| 18 | Comorbidities categoric | Nominal | Haven't (19,670) |
| | | | One (3,512) |
| | | | Two (757) |
| | | | Three or more (158) |
| 19 | Time for outcome | Numeric | 7.63 ± 6.83 |
| 20 | Hematology | Nominal | Haven't (23,649) |
| | | | Have (448) |
| 21 | Neurology | Nominal | Haven't (22,397) |
| | | | Have (1,700) |
| 22 | Oncology | Nominal | Haven't (24,048) |
| | | | Have (49) |
| 23 | Down syndrome | Nominal | Haven't (23,681) |
| | | | Have (416) |
| 24 | Nosocomial | Nominal | Haven't (23,476) |
| | | | Have (621) |

assessment, many of these factors could be excluded from our machine learning models without compromising predictive accuracy. This demonstrates the potential of simplifying mortality prediction while maintaining effective outcomes.

## Assessment of the developed models

In this study, COVID-19 mortality prediction models were developed using 12 ML algorithms, namely, GBC, ADA, CatBoost, RF, XGBoost, ET, LR, LDA, DT, NB, KNN, and QDA. These models were trained on three feature datasets: Dataset 1, containing all features; Dataset 2, with features selected by pediatricians; and Dataset 3, with features selected by the chi-squared independence test. The performance evaluation metrics used were accuracy, AUC, recall, precision, and sensitivity. The results are shown in Fig. 1.

In general, most of the models demonstrated comparable levels of accuracy, displaying good to excellent performance across all three datasets. More specifically, numerically, the models performed best when trained on Dataset 3, which was selected using the chi-square method, followed by Datasets 2 and 1. However, Dataset 1 still exhibited commendable performance even when all features were included. For Dataset 3, the highest accuracies were achieved by LR (92.53%), GBC (92.34%), and ADA (92.19%). For Dataset 2, GB (92.08%), ADA (91.92%), and LR (91.73%) achieved the highest accuracy. For Dataset 1,

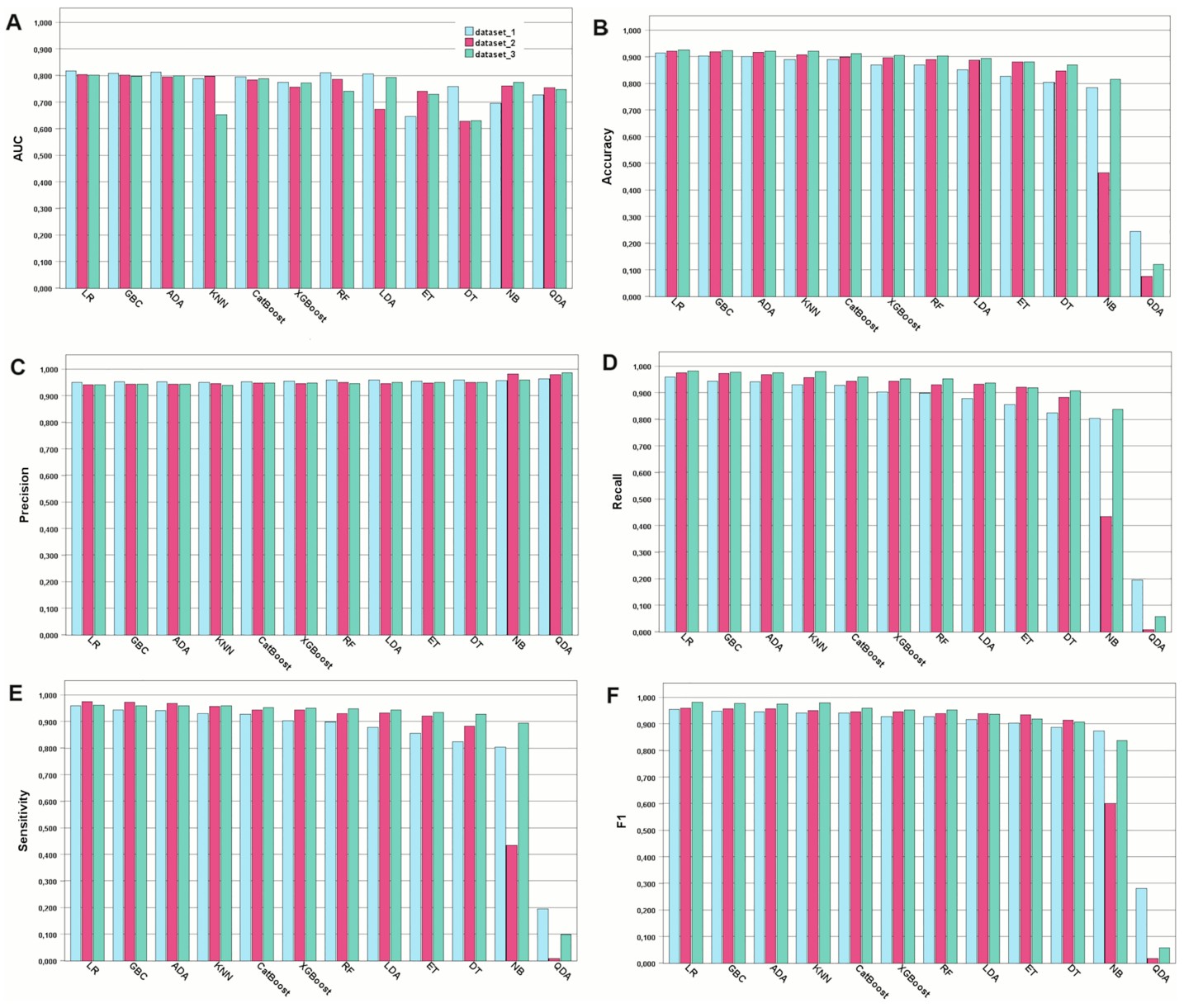

**Figure 1 Performance of the developed models for the metrics chosen.** Model performance for each type of metric. (A) Model performance with the AUC metric. (B) Accuracy of the developed models. (C) Precision metric for the developed models. (D) Recall metric. (E) Sensitivity metric. (F) F1-score-the harmonic mean between recall and precision               

GBC (91.41%), ADA (90.32%), and CatBoost (90.01%) were the best-performing models in terms of accuracy. Among the 12 algorithms analyzed, QDA consistently displayed the lowest performance across all datasets. Detailed comparison of the AUC for the top three models trained on Dataset 3, which achieved better results, is provided in Fig. 2. Considering the reliability of the AUC metric for imbalanced datasets, particularly relevant in our study despite using SMOTE for balancing, is crucial. The AUC results are nearly identical across all three datasets, with a notable emphasis on Dataset 1 containing all features.

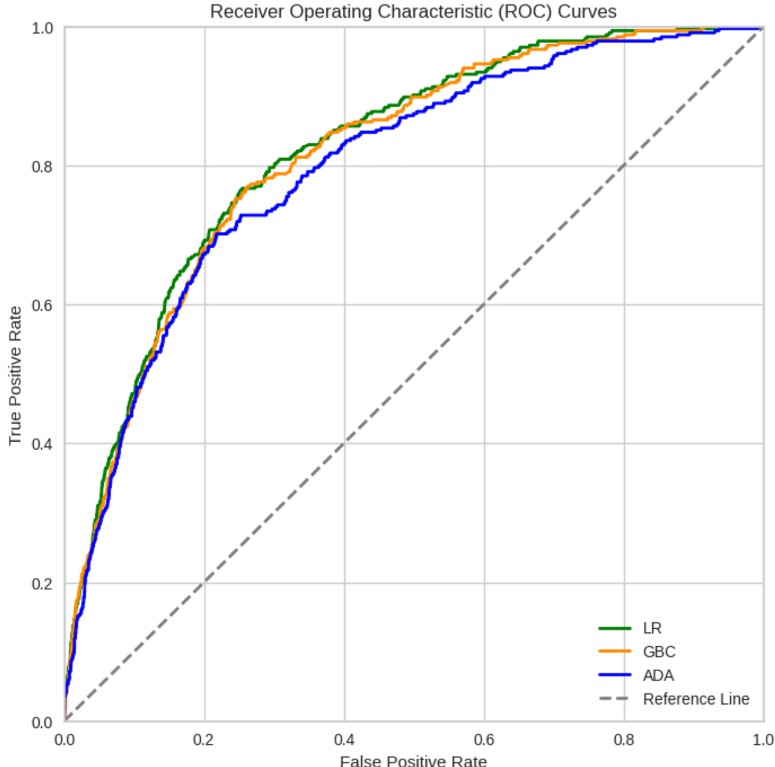

**Figure 2** ROC curves of the three best ML models for Dataset 3 that achieved better results.

## Model interpretation

Figure 3 presents a SHAP summary plot that visualizes the impact of each feature on the model's predictions for individual data points. Each line represents a data point, with the points distributed along the feature axis indicating their corresponding values. A wider spread of points for a given feature suggests a stronger influence on the model's output. Among the features, "oxygen saturation reduced" demonstrates the most significant impact on predictions. Blue points, representing normal oxygen saturation levels, are associated with favorable outcomes (patient discharge), whereas red points (low oxygen saturation) correlate with unfavorable outcomes (death). The comorbidity variable (categorical) also had a notable influence. Higher values of this feature, indicating a greater number of comorbidities (ranging from 0 to ≥ 3), are linked to an increased likelihood of predicting death. Similar trends were observed for "dyspnea", "respiratory distress", "total comorbidities", and "comorbidities".

## DISCUSSION

### Key points

This study aimed to develop and evaluate machine learning (ML) models for predicting COVID-19 mortality risk in Brazilian pediatric patients using a large public dataset. We analyzed demographic and clinical data to identify key mortality predictors. The ML

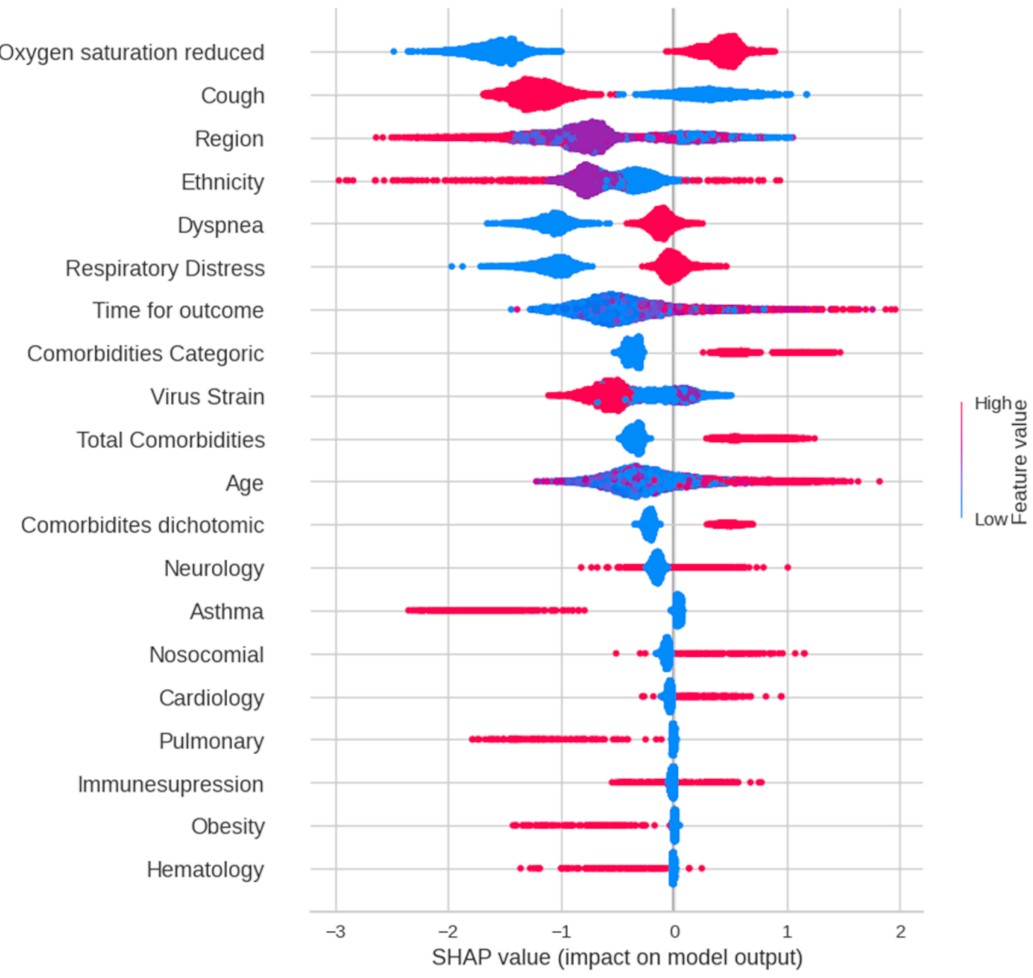

**Figure 3 A summary plot of SHAP values for mortality prediction on Dataset 3 (features selected by chi-squared test).** Blue dots indicate that low values in feature contribute to model to classify patient as discharge and red dots indicate high values of a feature contributes to model to classify patient as dead. More important features are in order top to bottom.

models were trained using three datasets: (i) all available features; (ii) features selected by pediatricians; and (iii) statistically relevant features. Although all models demonstrated robustness, our findings suggest that feature selection significantly enhances model performance. The model trained on the statistically relevant feature set (Dataset 3, 24 features) achieved the highest accuracy, followed by the model trained on pediatrician-selected features (Dataset 2, 17 features). The model using all features (Dataset 1) showed lower performance and may not generalize to other datasets. Our results indicate that simpler models with fewer features, such as those based on datasets 3 and 2, are preferable for clinical use as they require less input while maintaining high predictive accuracy. Consistent across all models, older age, low initial oxygen saturation, and pre-existing chronic conditions emerged as the strongest predictors of COVID-19 mortality in children and adolescents.

## Comparative analysis

We evaluated 12 ML algorithms for predicting mortality in hospitalized pediatric COVID-19 patients. LR demonstrated superior performance, achieving 92.5% accuracy, 98.11% sensitivity, 94.13% precision, 96.07% F1-score, and 80.15% AUC. GBC and ADA also yielded strong results (AUC ≥ 79.6%). Other models showed acceptable performance (AUC 80.1–81.6%), while DT and quadratic discriminant analysis (QDA) exhibited weaker results (AUC = 62.9%, accuracy 7.9–24.3%). To identify key predictors, we employed the XGBoost, random forest, and chi-squared tests. SHAP analysis revealed that reduced oxygen saturation, comorbidities, and older age were the most critical factors. These, along with 23 additional statistically significant features, enhanced ML model performance. Our findings align with previous studies that have reported some important clinical predictors for COVID-19 patient mortality, the most relevant features included age (*Moulaei et al., 2021, 2022*; *Wu et al., 2020*; *Yadaw et al., 2020*; *Zakariaee et al., 2023a, 2023b*), ethnicity (*Baqui et al., 2020, 2021*), geographic region (*Baqui et al., 2020, 2021*), dyspnea (*Shi et al., 2020*), cough (*Assaf et al., 2020*; *Das, Mishra & Saraswathy Gopalan, 2020*; *Gao et al., 2020*; *Moulaei et al., 2021, 2022*; *Zakariaee et al., 2023a, 2023b*), reduced oxygen saturation (*Assaf et al., 2020*; *Banoei et al., 2021*; *Kar et al., 2021*), cardiology disease (*Allenbach et al., 2020*; *Assaf et al., 2020*; *Baqui et al., 2020, 2021*; *Das, Mishra & Saraswathy Gopalan, 2020*; *Yadaw et al., 2020*; *Zakariaee et al., 2023a, 2023b*), pulmonary disease (*Banoei et al., 2021*; *Zakariaee et al., 2023a, 2023b*), immunosuppression (*Baqui et al., 2020, 2021*; *Gao et al., 2021*; *Xu et al., 2021*), renal (*Baqui et al., 2020, 2021*; *Shi et al., 2020*), asthma (*Aktar et al., 2021*; *An et al., 2020*; *Chimbunde et al., 2023*), total comorbidities (*Aktar et al., 2021*; *Banoei et al., 2021*), hematology disease (*Huyut, Velichko & Belyaev, 2022*; *Kamel et al., 2023*), neurology disease (*Baqui et al., 2020, 2021*; *Moulaei et al., 2021, 2022*; *Zakariaee et al., 2023a, 2023b*), oncology disease (*Assaf et al., 2020*; *Chin et al., 2020*; *Hu, Yao & Qiu, 2020*; *Zakariaee et al., 2023a, 2023b*), hypertension (*Assaf et al., 2020*; *Das, Mishra & Saraswathy Gopalan, 2020*; *Yadaw et al., 2020*; *Zakariaee et al., 2023a, 2023b*), and chromosomal abnormalities (*Landes et al., 2021*).

Below, we detail how each of these best models selected on tests works technically.

### Logistic regression

Logistic regression is a statistical model used for binary classification, where the output *YY* takes values *00* or *11*. The model is based on the logistic (sigmoid) function, defined as:

$$P(Y = 1|X) = \sigma(z) = \frac{1}{1 + e^{-z}}$$

where:

- $\sigma(z)$ is the sigmoid function, which compresses output values to the range $(0, 1)$.
- $z$ is the linear combination of predictors:
- $\beta 0$ are the coefficients of the predictors.

- Logistic regression is trained by maximizing the log-likelihood function:

$$L(\beta) = i = 1 \sum^m [yi \log Pi + (1 - yi) \log(1 - Pi)]$$

where:

- $yi$ are the actual labels (0 or 1).
- $Pi$ are the probabilities predicted by the model.
- $m$ is the total number of observations.

## Gradient boosting classifier

Gradient boosting is a machine learning method based on sequential decision trees. It minimizes a loss function using gradient descent.

Gradient boosting steps:

- Start with an initial prediction, typically the mean of the target values.
- Train a decision tree $ht(X)$ to minimize the residuals of the previous prediction.

$$Rt = Y - Ft - 1(X)$$

- The new prediction is updated by adding the weighted tree output:

$$Ft(X) = Ft - 1(X) + \eta ht(X)$$

- Repeat the process until a stopping criterion is met, such as a maximum number of iterations or minimum error.

The loss function depends on the task:

- Binary classification: Log-loss

$$L(y, \hat{y}) = -\sum_{i=1}^m [y_i \log \hat{y}_i + (1 - y_i) \log(1 - \hat{y}_i)]$$

- Regression: mean squared error (MSE)

$$L(y, \hat{y}) = \frac{1}{m} \sum_{i=1}^m (y_i - \hat{y}_i)^2$$

Popular GBC-based models include XGBoost, LightGBM, and CatBoost, widely used in machine learning applications for structured data.

### Adaptive boosting

AdaBoost is an ensemble learning method that combines multiple weak classifiers (typically shallow decision trees) to create a strong classifier.

AdaBoost steps:

Each sample $(i)$ is assigned a weight $w_i$, initially equal for all samples:

$$w_i = \frac{1}{m}.$$

Train a weak classifier $h_t(x)$, minimizing the weighted error $\varepsilon_t$:

$$\varepsilon_t = \sum_{i=1}^{m} w_i I(h_t(x_i) \neq y_i).$$

Compute a classifier importance coefficient based on $\varepsilon_t$:

$$\alpha_t = \frac{1}{2}\log\left(\frac{1 - \varepsilon_t}{\varepsilon_t}\right).$$

Update sample weights, giving higher weight to misclassified samples:

$$w_i^{(t+1)} = w_i^{(t)} \cdot e^{\alpha_t I(h_t(x_i) \neq y_i)}.$$

Repeat until a maximum number of classifiers is reached or error is minimized. The final prediction is given by a weighted sum of individual classifiers:

$$H(x) = \text{sign}\left(\sum_{t=1}^{T} \alpha_t h_t(x)\right).$$

AdaBoost enhances accuracy by reducing errors from weak classifiers and is widely used for classification tasks in structured datasets.

Few studies have explored the use of ML models to predict mortality in children and adolescents with COVID-19. We recently conducted a systematic review of clinical prediction models developed using supervised ML algorithms for this population (*Dos Santos et al., 2024*). Our analysis included ten studies of six focused on diagnosis and four on prognosis. All models predicted binary outcomes with disease detection being the most common target. Tree-based and neural-network models were the predominant ML techniques employed. However, most studies suffered from limitations, including small sample sizes, inconsistent reporting, potential data biases, and inadequate reporting of essential metrics, such as calibration, discrimination, and hyperparameters. These deficiencies hinder the reproducibility and limit the generalizability of their findings. While ML models have been applied to various pediatric outcomes beyond COVID-19, the evidence base for predicting mortality in COVID-19 remains scarce and characterized by methodological shortcomings.

Although some studies have utilized artificial intelligence as a tool to predict clinical outcomes in children, we identified only one study that employed machine learning methodology to predict outcomes in children with COVID-19. *Gao et al. (2022)* proposed a machine learning model (MedM) and evaluated its performance in predicting hospitalization and disease severity in a pediatric population with confirmed COVID-19. Based on electronic health records, MedML extracted the most predictive features based on medical knowledge and propensity scores from over six million medical concepts and incorporated the inter-feature relationships in medical knowledge graphs *via* graph neural networks. Subsequently, the researchers evaluated MedML on the National Cohort Collaborative (N3C) dataset and found that it achieved up to a 7% higher AUROC and 14% higher AUPRC than the best baseline machine-learning models. However, the AUCROC performance of the models ranged from 0.62 (DT) to 0.75 (MedML), whereas in

our analysis the AUCROC was approximately 0.80 for almost all models evaluated. It is important to note that the variables used to develop the models were quite different, as we utilized a specific database that gathered data on COVID-19, whereas *Gao et al. (2022)* used an administrative database with very different covariables.

ML models are often characterized in the literature as black boxes, lacking transparency in how individual features contribute to their predictions (*Reddy, 2022*). Given that these models make decisions based on specific feature values, understanding these decisions is crucial, especially in critical domains, such as healthcare, where patient well-being is at stake (*Rasheed et al., 2022*). Explainable Artificial Intelligence (XAI) (*Lundberg et al., 2020*) addresses this challenge by enhancing model interpretability and trustworthiness. One prominent XAI method is the Shapley Additive Explanation (SHAP) value, which decomposes model outputs into feature-based contributions. Based on cooperative game theory, SHAP provides a comprehensive measure of feature importance by considering all possible feature combinations. Shapley value is a common measurement of individual feature importance and is widely utilized in the interpretability analysis of machine learning models (*Hyland et al., 2020*; *Winter, 2022*). A positive Shapley value indicates that the feature is positively correlated with the target of interest, and higher values suggest higher importance, whereas negative Shapley values correspond to negative correlations. As a *post-hoc* technique, SHAP is applicable to any machine-learning model. In this study, we leveraged SHAP to improve the interpretability of our GBC model by elucidating the influence of features such as reduced oxygen saturation, comorbidities (represented as numerical, binary, or ordinal variables), dyspnea, and respiratory distress at admission as reliable predictors of mortality in pediatric patients with COVID-19. It is noteworthy that in the study of *Gao et al. (2022)*, according to the Shapley values, the indicators for severity prediction task are larger for BMI, creatinine, and glucose. We posit that these markedly different indicators between the studies strongly illustrate the impact of dataset selection in developing clinical prediction models on the performance and applicability of the models.

## Machine learning models and larges scale datasets

The machine learning algorithms presented in our work have proven to be feasible for large databases. Specifically, models such as logistic regression, AdaBoost, and gradient boosting demonstrate significant utility due to their inherent scalability and computational efficiency relative to their predictive power. Logistic regression, particularly when implemented with stochastic gradient descent (SGD) or its variants, can process vast amounts of data incrementally without requiring the entire dataset to reside in memory, making it suitable for streaming or disk-based learning scenarios. Ensemble methods like AdaBoost and gradient boosting, while potentially more computationally intensive per iteration, derive their strength from building sequences of weak learners; this iterative nature allows for potential parallelization and, crucially, they often achieve high accuracy with relatively shallow trees as base learners, mitigating the complexity explosion sometimes seen in other non-linear methods. Their proven effectiveness across diverse large-scale benchmarks underscores their suitability for modern data-rich environments (*Hastie, Tibshirani & Friedman, 2009*).

Furthermore, the specific characteristics of these algorithms lend themselves well to the challenges posed by large databases, such as high dimensionality and the presence of complex, non-linear relationships. Logistic regression, often combined with L1 or L2 regularization, provides a robust linear baseline capable of handling sparse, high-dimensional feature spaces commonly encountered in large datasets, while also offering interpretable coefficients. Boosting algorithms, particularly gradient boosting, excel at capturing intricate patterns and interactions within the data by sequentially fitting models to the residuals of prior iterations. This adaptive fitting process allows them to model complex functions effectively without necessarily overfitting, especially when parameters like learning rate, tree depth, and subsampling are carefully tuned. The capacity of gradient boosting frameworks to optimize arbitrary differentiable loss functions further enhances their flexibility for diverse large-scale prediction tasks (*Friedman, 2001*).

Contrasting these machine learning approaches with traditional statistical methods reveals critical differences when applied to large databases. While traditional methods, such as ordinary least squares regression or maximum likelihood estimation for generalized linear models, provide rigorous inferential frameworks, they often rely on assumptions (*e.g.*, normality of errors, specific distributional forms) that may be violated in massive, heterogeneous datasets. Moreover, many classical techniques involve computations, like matrix inversion, that scale poorly with the number of samples or features, rendering them computationally infeasible for very large n or p without specialized implementations. Machine learning models, particularly those discussed, often prioritize predictive accuracy and computational scalability, frequently employing iterative optimization techniques and making fewer stringent assumptions about data generation processes. This distinction, emphasizing prediction over parameter inference or exact distributional modeling, aligns well with the practical demands of extracting actionable insights from large-scale data repositories (*Breiman, 2001*).

## Strengths of the study

This study leveraged a nationwide database to provide a comprehensive overview of COVID-19 in Brazilian pediatric patients. The large sample size of laboratory-confirmed cases enabled the rigorous evaluation of multiple ML algorithms. Our findings suggest that ML models are robust when applied to extensive specifically designed datasets, offering the potential for future public health applications.

## Limitations of the study

This study has several limitations. First, the SIVEP-Gripe database, which focuses on hospitalized patients, restricts the generalizability of the findings to a broader pediatric population. Second, we were unable to conduct an external validation of our models, an important step in the development of clinical prediction models (*Steyerberg & Harrell, 2016*). This involves evaluating the performance of a model on an independent dataset that was not used during the model's training or internal validation phases (*Collins et al., 2024a*; *Collins & Moons, 2019*; *Collins et al., 2024b*). This process addresses critical issues such as

overfitting and bias, while ensuring that the model generalizes well to new, unseen data, which is essential for its real-world applicability. Nevertheless, many of the available datasets are too small to provide reliable answers (*Riley et al., 2024a*, *2024b*). To address this pivotal issue, we are currently integrating data from all official Brazilian databases, including non-hospitalized and hospitalized patients, across the country. Therefore, we believe that with this updated dataset, encompassing more than two million pediatric cases, we will be able to use some modern recommended techniques, such as the use of resampling methods for internal validation, to evaluate model performance and generalizability across clusters. Third, the administrative nature of the database hinders the assessment of certain clinical management details. Additionally, missing data, a common challenge in such registries, was mitigated through meticulous manual review of case records, including in-depth analysis of the "clinical observation" field in SIVEP-Gripe database. Finally, the absence of a national audit system for the SIVEP Gripe database is a notable limitation. However, our extensive analysis of these data since the onset of the pandemic has yielded consistent results comparable to those from other low-to middle-income countries using conventional statistical techniques (*Nachega et al., 2022*).

## Clinical and policy implications

We believe that the utilization of ML models to predict outcomes in pediatric COVID-19 cases and other public health threats presents a transformative potential for healthcare systems. The application of ML models in predicting mortality in children with COVID-19 may have significant clinical, research, and policy implications. Our findings indicate that ML models can assist in accurately identifying high-risk children at an early stage, enabling healthcare systems to allocate resources (*e.g.*, ICU beds, ventilators, and medications) more effectively for those at the highest risk of severe outcomes. ML models can inform individualized treatment plans by identifying risk factors specific to pediatric populations, potentially leading to the development of tailored clinical guidelines (*Chumachenko et al., 2024*; *Collins et al., 2024b*; *Singh, 2019*; *Wynants et al., 2020*). Predictive models can help policymakers prioritize vaccination strategies for children at elevated risk of severe outcomes, particularly in resource-limited settings. Furthermore, ML models can be integrated into public health surveillance systems to monitor trends in pediatric COVID-19 mortality and to inform targeted interventions (*Bragazzi et al., 2020*). However, policymakers must address ethical, equity, and regulatory challenges to ensure the effective implementation of these tools. Collaboration among researchers, clinicians, and policymakers is essential to maximize the benefits of ML in pediatric care (*Malhotra et al., 2023*).

## CONCLUSIONS

In summary, this study evaluated the performance of various ML algorithms in predicting mortality among hospitalized pediatric COVID-19 patients. LR, GBC, and ADA models demonstrated superior performance in accurately identifying patients at risk of death, offering potential benefits for resource allocation and patient outcomes. Our findings underscore the critical role of factors such as low oxygen saturation and comorbidities in

predicting mortality. An LR model incorporating these predictors effectively identified high-risk patients on admission. The application of ML models could streamline decision making in clinical and public health settings, potentially improving survival rates. Further research is needed to explore additional predictors and evaluate the long-term impact of COVID-19 in pediatric patients.

### Funding
This research was funded by the CNPq (Conselho Nacional de Desenvolvimento Científico e Tecnológico, grant number 405731/2023-0) and FAPEMIG (Fundação de Amparo à Pesquisa do Estado de Minas Gerais, grants number APQ-03205-24 and APQ-04839-24). The APC was funded by FAPEMIG (Fundação de Amparo à Pesquisa do Estado de Minas Gerais, grant APQ-03205-24). The funders had no role in study design, data collection and analysis, decision to publish, or preparation of the manuscript.

### Grant Disclosures
The following grant information was disclosed by the authors:
CNPq (Conselho Nacional de Desenvolvimento Científico e Tecnológico): 405731/2023-0.
FAPEMIG (Fundação de Amparo à Pesquisa do Estado de Minas Gerais): APQ-03205-24, APQ-04839-24 and APQ-03205-24.

### Competing Interests
The authors declare that they have no competing interests.

### Author Contributions
- Adriano Lages dos Santos conceived and designed the experiments, performed the experiments, analyzed the data, performed the computation work, prepared figures and/or tables, authored or reviewed drafts of the article, and approved the final draft.
- Maria Christina L. Oliveira analyzed the data, prepared figures and/or tables, authored or reviewed drafts of the article, and approved the final draft.
- Enrico A. Colosimo performed the experiments, analyzed the data, performed the computation work, prepared figures and/or tables, authored or reviewed drafts of the article, and approved the final draft.
- Robert H. Mak analyzed the data, prepared figures and/or tables, authored or reviewed drafts of the article, and approved the final draft.
- Clara C. Pinhati analyzed the data, prepared figures and/or tables, authored or reviewed drafts of the article, and approved the final draft.
- Stella C. Gallante analyzed the data, prepared figures and/or tables, authored or reviewed drafts of the article, and approved the final draft.
- Hercílio Martelli-Júnior analyzed the data, prepared figures and/or tables, authored or reviewed drafts of the article, and approved the final draft.

- Ana Cristina Simões e Silva conceived and designed the experiments, analyzed the data, performed the computation work, prepared figures and/or tables, authored or reviewed drafts of the article, and approved the final draft.
- Eduardo A. Oliveira conceived and designed the experiments, performed the experiments, analyzed the data, performed the computation work, prepared figures and/or tables, authored or reviewed drafts of the article, and approved the final draft.

### Ethics

The following information was supplied relating to ethical approvals (*i.e.*, approving body and any reference numbers):

The study was approved by the Federal University of Minas Gerais institutional review board (register 6.127.414). The funding organizations had no role in the design and conduct of the study; collection, management, analysis, and interpretation of the data; and preparation, review, or approval of the manuscript.

### Data Availability

The SIVEP-Gripe data are available at: https://opendatasus.saude.gov.br/dataset.

The source code and data used in the project is available at GitHub and Zenodo:

- https://github.com/adrianocomp/sivep_phd.

- Adriano Santos. (2025). adrianocomp/sivep_phd: doi (doi). Zenodo. https://doi.org/10.5281/zenodo.15350456.

### Supplemental Information

Supplemental information for this article can be found online at http://dx.doi.org/10.7717/peerj-cs.2916#supplemental-information.

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
