# Peer review of "Comparative performance of twelve machine learning models in predicting COVID-19 mortality risk in children: a population-based retrospective cohort study in Brazil"

_PeerJ Computer Science, doi:10.7717/peerj-cs.2916_

## Round 0.1 · original submission · Major Revisions

The work has several shortcomings that require a thorough review before it can be considered for publication. The introduction lacks an adequate literature review, as it does not include references to recent studies or pioneering work in the field, nor does it provide a clear organization of the article. Furthermore, the relevance of the study and the choice of the Brazilian SIVEP-Gripe dataset are not adequately justified. On the experimental level, several critical issues arise related to the choice of machine learning algorithms, which is not adequately motivated compared to other techniques, such as deep learning architectures or alternative methods. There are doubts about the generalizability of the models, particularly with respect to the risk of bias in the dataset and the representativeness of the pediatric population. The data preparation also raises concerns, including issues of completeness, noise, and diversity. Regarding the validity of the results, there is a lack of comparison with the state of the art and a clear demonstration of the claims made, such as the one on the feasibility of the models for large datasets, which is not supported in the results section. The results themselves are not adequately linked to the abstract, and the clinical relevance or practicality of the proposed models is not clearly evident. Finally, the quality of the presentation is improvable, with a need to standardize the bibliographic sources and to better justify the choice of metrics used. Overall, the work has the potential to make a significant contribution, but requires substantial revision to address methodological issues, improve the interpretability of the models and ensure adequate comparison with similar studies.

·

Basic reporting

1)The 'Introduction' section contains almost no literature review. This section should be developed by providing more detailed information about the study, and should be addressed by referencing relevant studies and including information about the datasets, methods and results of these studies. Particular attention should be paid to referencing pioneering publications in the field and publications made in the last 5 years.

Experimental design

1) Information about evaluation methods (at least the most successful ones, such as Logistic regression (LR), gradient boosting (GBC), and adaptive boosting (ADA) models) should be provided.
2) Formulas for assessment metrics can be given with references.

Validity of the findings

The results and limitations are explained. However, the comparisons were made only within the study. Therefore, the lack of literature review emphasized in the first section should be eliminated and a table containing comparisons with similar studies should be added in the 'Comparative analysis' section.

Additional comments

It is thought that if the necessary revisions are made carefully, they will contribute to the field.

·

Basic reporting

1. In international studies, language differences in data or medical terms across regions can lead to misinterpretation or inconsistencies in the dataset.
2. The overall organization of the paper is not presented in the introduction.

Experimental design

1. If the dataset used is incomplete, noisy, or contains errors, the machine learning model will perform poorly. Missing data and inconsistencies in data entry can also harm predictive power.
2. Small sample sizes in specialized populations (like pediatric patients) lead to a lack of generalizability. Overfitting becomes a risk when the sample size is too small, leading to poor performance on unseen data.
3. If the data reflects any form of bias (for example, socioeconomic, geographical, or racial bias), the model may be skewed in predicting outcomes, leading to unfair or incorrect decisions.
4. Machine learning models require data that represent diverse populations. A lack of this diversity can affect the performance when the model is applied to real-world or global populations.
5. Overfitting occurs when a model is too tailored to the training data, leading to high accuracy on the training set but poor performance on new, unseen data.
6. Failing to identify and select the most important features can result in a less accurate model, as the algorithm might rely on irrelevant data or overlook key indicators.
7. Complex ML models like deep learning may perform well but can be difficult for clinicians to interpret, reducing their trust and practical utility in decision-making.
8. Inconsistent formatting and scaling of data (e.g., age, blood pressure, or oxygen levels) can affect model accuracy and comparison, especially in heterogeneous datasets.
9. In international studies, language differences in data or medical terms across regions can lead to misinterpretation or inconsistencies in the dataset.
10. The data may be time-sensitive, such as datasets based on older versions of treatment protocols, and may fail to reflect recent advancements in healthcare, leading to outdated predictions.
11. In healthcare, properly labeling patient data for outcomes like mortality can be subjective or inconsistent, which can distort machine learning model performance.
12. Some machine learning models might not scale well with large datasets or may be too computationally expensive, making them impractical for real-world applications with vast datasets.
13. A model trained on data from one population (e.g., specific to Brazil) might not generalize well to other populations, particularly due to differences in healthcare practices or underlying diseases.
14. Insufficient validation or testing on unseen data can make the model's true effectiveness unclear. Cross-validation is necessary but often overlooked.
15. There’s often a trade-off between model complexity (which typically improves performance) and interpretability (which may suffer as the model gets more complex).
16. Machine learning algorithms sometimes struggle with processing categorical variables (e.g., race, ethnic groups, gender) effectively, leading to information loss or inaccuracies.
17. Machine learning systems may fail to adequately handle errors or inconsistencies in input data, which can lead to system crashes or poor predictions when encountering unexpected data types or anomalies.
18. In clinical applications, ML models might provide predictions, but without clear explanations (e.g., decision trees or feature importance), they could be impractical for clinicians to use in decision-making.
19. Even with high accuracy in test settings, the model may fail to generalize well to different types of cases or subgroups within the target population.

Validity of the findings

1. If machine learning models are heavily reliant on historical data, they may fail to adapt to changes in disease progression or treatment methods, reducing their relevance over time.
2. In healthcare, privacy is a major concern, especially when dealing with sensitive data such as health records. If not addressed, privacy risks could violate legal standards (e.g., HIPAA) or trust in the system.
3. Some machine learning algorithms, particularly deep learning, require significant computational power, time, and resources to train, which can be limiting for healthcare organizations with fewer resources.
4. If the model does not include sufficient feedback loops from clinicians or stakeholders, it may not continue to improve, missing opportunities for optimizing its predictions based on real-world inputs.
5. Models may present predictions as definite outcomes, even when there is significant uncertainty, leading to false confidence or misinformed decision-making.
6. Data breaches and security vulnerabilities are critical concerns, particularly in healthcare contexts where malicious attacks or inadvertent leaks of sensitive patient data could result in serious consequences for both patients and institutions.

Reviewer 3 ·

Basic reporting

The document is well-written and the language is adequate. However, there are three issues that must be addressed in the introduction.
• Justify the use of machine learning algorithms as opposed to alternative methods such as Principal Component analysis
• According to UNICEF, COVID-19 deaths in children are the lowest among group ages. Therefore, the authors need to justify the relevance of study. See: https://data.unicef.org/topic/child-survival/covid-19/
• The introduction can be improved by describing the characteristics of the children in Brazil and therefore justifying the selection of data from that country.

Experimental design

The article is within the scope of the journal, and a rigorous investigation was done. While the methods are well-established algorithms, it would be beneficial to provide a brief description of each selected algorithm. The data preparation was adequate, and the evaluation metrics employed and align with other studies. The style of the sources need to be uniformed, the style of source 1,2 and 3 are different.

Validity of the findings

The algorithms implemented were validated properly.
Before publication, the authors need to clarify how their findings are relevant to research, clinical practice or public policy decisions.

Additional comments

The article can be published after the previous comments are addressed.

Reviewer 4 ·

Basic reporting

In my opinion, this article has many shortcomings and needs to address the following comments:
1. The authors make use of 12 ML models for predicting the outcomes without justifying the reasons for choosing these 12 algorithms. There are many powerful DL architectures covering neural networks which will do the job more effectively. I suggest authors to explain how the selection of these 12 ML algorithms turned effective for current research.
2. This research lacks in providing enough background for addressing the challenges of mortality prediction in pediatric populations. I suggest authors to make clear distinction how mortality rates differ in different age groups.
3. The authors use AUC, accuracy, precision, and recall as the performance metrics for evaluating this research. It will be interesting if authors will reveal how these metrics are significant in pediatric data predictions.
4. The authors use SIVEP-Gripe dataset, which is specific to Brazil for training the models without addressing the concerns of bias in the dataset. I suggest authors to address this issue.
5. This research claims "ML models are feasible for large-scale datasets". I suggest author to prove this claim of feasibility in your result section. Note there is a gap in the abstract and the result section.
6. This research must compare the evaluated results with state-of-the-art and prove how current research is outperforming the existing studies.

Experimental design

The experimental design is weak in this research. The authors choose to develop ML models for regional dataset without revealing their significance.

Validity of the findings

Finding are weak. This research must compare the evaluated results with state-of-the-art and prove how current research is outperforming the existing studies.

---

## Round 0.2 · Minor Revisions

We thank the authors for the review work already done and for the changes made. After analyzing the new reviews, a contrast emerges between the two opinions received: while the first reviewer considers the issues resolved and the manuscript acceptable, the second believes that some key aspects, especially regarding the experimental setup and the validity of the dataset, have not been adequately addressed.

In particular, the authors are asked to:

- Strengthen the experimental section by addressing the critical issues highlighted, so as not to limit themselves to an integration in the limitations section.

- Clarify and better justify the claim on the feasibility of machine learning models for large datasets, avoiding a simple removal of the reference but rather arguing the context and limitations of the analysis conducted.

- Improve the consistency between the abstract and the results section, as highlighted by the second reviewer.

These changes should help to strengthen the quality and robustness of the work. Once implemented, the manuscript may be considered for publication.

Reviewer 3 ·

Basic reporting

My previous comments have been addressed in the new version of the manuscript

Experimental design

The new manuscript has addressed my previous concerns

Validity of the findings

the findings correspond to the data analysis

Additional comments

This version of the manuscript includes answers to the all the reviewer's concerns

Reviewer 4 ·

Basic reporting

Most of the comments provided in the first revision are not addressed. There is not enough improvement in the revised version.

Experimental design

I tried to reveal the core issues in the experimental set-up along with the dataset used in this article. Instead of revising the dataset along with experiments, authors are claiming to have added points in the limitations section.

Validity of the findings

5. This research claims "ML models are feasible for large-scale datasets". I suggest author to prove this claim of feasibility in your result section. Note there is a gap in the abstract and the result section.
Answer: We acknowledge your valuable comment and agree with your observations. Consequently, we removed these statements from the Results section. It is anticipated that with the current availability of large-scale datasets and advanced computational capabilities, this pertinent question will be thoroughly investigated to determine the comparative advantages and limitations of conventional statistical analyses and artificial intelligence methodologies.

- Your entire study is based on this theme. Do you think deleting these details from your result section will improve this article?
- When this pertinent question will be thoroughly investigated to determine the comparative advantages and limitations of conventional statistical analyses and artificial intelligence methodologies?

---

## Round 0.3 · accepted · Accept

All comments of the reviewers have been addressed.

Reviewer 3 ·

Basic reporting

No comment

Experimental design

The current version has improved the metholodogy section

Validity of the findings

No comment

Additional comments

This version has addressed the reviewer's concerns and now it is clear the motivation of the study and the motivation for using ML algorithms. Furthermore, the description of the methodology has improved

Reviewer 4 ·

Basic reporting

The revised version addresses all my comments and looks improved. I have no further comments.

Experimental design

All comments are addressed

Validity of the findings

All comments are addressed